# Hydroxyprolyl-Glycine in 24 H Urine Shows Higher Correlation with Meat Consumption than Prolyl-Hydroxyproline, a Major Collagen Peptide in Urine and Blood

**DOI:** 10.3390/nu16203574

**Published:** 2024-10-21

**Authors:** Tomoko T. Asai, Satoshi Miyauchi, Sri Wijanarti, Ayaka Sekino, Akiko Suzuki, Sachiko Maruya, Takayo Mannari, Ai Tsuji, Kenji Toyama, Rieko Nakata, Yasunori Ogura, Hitoshi Takamura, Kenji Sato, Ribeka Takachi, Satoru Matsuda

**Affiliations:** 1Department of Food Science and Nutrition, Faculty of Human Life and Environment, Nara Women’s University, Nara 630 8506, Japan; 2Division of Applied Biosciences, Graduate School of Agriculture, Kyoto University, Kitashirakawa Oiwake-cho, Sakyo-ku, Kyoto 606 8502, Japan; 3Kyousei Science Center for Life and Nature, Nara Women’s University, Kitauoya-Nishimachi, Nara 630 8506, Japan

**Keywords:** prolyl-hydroxyproline, hydroxyprolyl-glycine, collagen peptide, meat, biomarker

## Abstract

**Background.** Urinary collagen peptides, the breakdown products of endogenous collagen, have been used as biomarkers for various diseases. These non-invasive biomarkers are easily measured via mass spectrometry, aiding in diagnostics and therapy effectiveness. **Objectives.** The objective of this study was to investigate the effects of consuming collagen-containing meat on collagen peptide composition in human blood and urine. **Methods.** Ten collagen peptides in 24 h urine were quantified. **Results.** Prolyl-hydroxyproline (Pro-Hyp) was the most abundant peptide. Except for hydroxyprolyl-glycine (Hyp-Gly), levels of other minor collagen peptides showed high correlation coefficients with Pro-Hyp (r = 0.42 vs. r > 0.8). Notably, 24 h urinary Hyp-Gly showed a correlation coefficient of r = 0.72 with meat consumption, significantly higher than the coefficient for Pro-Hyp (r = 0.37). Additionally, the levels of Pro-Hyp and Hyp-Gly in the blood of seven young women participants increased similarly after consuming fish meat, while before ingestion, only negligible amounts of Hyp-Gly were present. To examine which peptides are generated by the degradation of endogenous collagen, mouse skin was cultured. The amount of Pro-Hyp released from the skin was approximately 1000-fold higher than that of Hyp-Gly. Following consumption of collagen-containing meat, both Pro-Hyp and Hyp-Gly are released in blood and excreted into urine, although Pro-Hyp is primarily generated from endogenous collagen even under physiological conditions. **Conclusions.** Therefore, in 24 h urine samples, the non-negligible fraction of Pro-Hyp is contributed by endogenous collagen, making 24 h urine Hyp-Gly level a potential biomarker for evaluating meat consumption on the day.

## 1. Introduction

Collagen, the main protein in the extracellular matrix of animals, has a triple-helical structure which is resistant to most endopeptidases, except for collagenases. When collagen is heated in water, the stable triple-helical structure collapses, forming a random coil structure and increasing its vulnerability to protease degradation. This denatured form of collagen (commonly referred to as gelatin) in food is degraded into peptides and amino acids by endo- and exopeptidases in the gastrointestinal tract [1,2]. In addition, food-grade protease digests of gelatin derived from the connective tissues of fish, birds, and animals are extensively produced on an industrial scale as food ingredients and marketed under various names, including gelatin hydrolysate, collagen hydrolysate [3], collagen peptide, or simply “collagen”, in markets. In the following sections, we referred to it as collagen hydrolysate.

Numerous human clinical trials, including randomized, double-blind, and placebo-controlled ones, have demonstrated that supplementation of collagen hydrolysate exerts beneficial effects on skin conditions, such as enhancement of skin moisture, elasticity, and epidermal barrier function and decreases in wrinkle volume [4,5]. It has also been shown to attenuate osteoarthritis symptoms [6] and improve the healing of pressure ulcers [7,8]. Some collagen hydrolysate-containing food products have been approved by the Japanese Consumer Affairs Agency as special dietary supplements for patients with pressure ulcers [9].

Supplementation with collagen hydrolysate can augment the levels of food-derived collagen peptides such as prolyl-hydroxyproline (Pro-Hyp), hydroxyprolyl-glycine (Hyp-Gly), and prolyl-hydroxyprolyl-glycine (Pro-Hyp-Gly) in human blood [10,11]. In vitro studies have shown that the collagen peptides Pro-Hyp and Hyp-Gly can enhance the proliferation of fibroblasts attached to collagen fibrils [12,13]. Furthermore, Pro-Hyp has been reported to increase production of glycosaminoglycans by fibroblasts [14] and chondrocytes [15]. These facts indicate that food-derived collagen peptides such as Pro-Hyp and Hyp-Gly are directly responsible for the beneficial effects observed from collagen hydrolysate supplementation. Additionally, animal experiments have demonstrated that collagen peptides are naturally produced in tissues undergoing inflammation [16] and at wound healing sites on the skin [17], as endogenous collagen breaks down into smaller peptides as part of the body’s response to tissue injury or inflammation. Inflammatory tissues primarily produced Pro-Hyp, while there was no discernible increase in Hyp-Gly. In patients with bone metastatic cancer, Pro-Hyp was also reported to increase in blood [18].

We have previously demonstrated that the levels of collagen peptides in human blood plasma are increased by both consumption of cooked fish meat containing collagen as well as supplementation with collagen hydrolysate [19]. Thus, collagen peptides are generated by the digestion and absorption of collagen-containing foods, which then enter the blood circulation.

Daily ingestion of collagen hydrolysates for 4 weeks altered the contents and structures of collagen peptides in the human blood [11], implying that frequent consumption of collagen-containing foods such as fish, chicken, and animal meat might affect the content and structure of collagen peptide in blood.

Both endogenous and food-derived collagen peptides have been demonstrated to be excreted in urine [20,21,22]. Since urine collection is a non-invasive technique compared to that of blood, urinary collagen peptides have a potential to serve as a marker of dietary collagen intake. However, the impact of the consumption of collagen-containing foods on the content and structure of collagen peptide in urine remains unclear. The objective of the present study was to elucidate the impact of the consumption of collagen-containing foods on contents and structures of collagen peptides in 24 h urine specimens and the underlying mechanism of this urinary release.

## 2. Materials and Methods

### 2.1. Materials

A standard mixture of amino acids (Type H), acetonitrile (HPLC-grade), trimethylamine, phenyl isothiocyanate (PITC), piperidine, 4-methylmorpholine, N,N-dimethylformamide, t-butyl methyl ether, trifluoroacetic acid, and isoflurane were purchased from Wako Pure Chemical Industries (Osaka, Japan). 9-Fluorenylmethoxycarbonyl (Fmoc) amino acid derivatives, Fmoc glycine-bond p-alkoxybenzyl alcohol (Alko) resin, proline- and hydroxyproline-bond 2-chlorotrityl chloride (Barlos) resin, 1H-benzotriazol-1-yloxy-tri-(pyrrolidino)phosphonium hexafluorophosphate (PyBOP), and 1-hydroxybenzotriazole (HOBt) were purchased from Watanabe Chemical Industries (Hiroshima, Japan). 6-Aminoquinolyl-N-hydrox-ysuccinimidyl carbamate (AccQ) was obtained from Toronto Research Chemicals (Toronto, ON, Canada). Pro-Hyp and Hyp-Gly were obtained from Bachem (Budendorf, Switzerland). L-Hydroxyproline (Hyp), Dulbecco’s phosphate-buffered saline (D-PBS), Dulbecco’s Modified Eagle’s Medium (DMEM) (1.0 g/L Glucose), and penicillin–streptomycin mixed solution were purchased from Nacalai Tesque (Kyoto, Japan). All other reagents were of analytical grade or better. Fetal bovine serum (FBS) was obtained from Biowest (Nuaille, France). Low-molecular weight compounds (<6000 Dalton) in the FBS, including collagen peptides, were removed using an Econo-Pac 10DG column (Bio-Rad Laboratories, Hercules, CA, USA) as described previously [23]. The FBS without the low-molecular weight compounds is referred to as FBS-FHP (free from hydroxyprolyl peptide).

### 2.2. Human Trial 1

The trial participants were women undergraduate students enrolled for a practical training in the registered dietician course at Nara Women’s University. This training course is conducted with the approval of the Faculty of Life and Environmental Science, Nara Women’s University. In this training, dietary intakes for the previous 1–2 months and on the day of investigation were examined, and 24 h urine samples were collected. The 24 h urine collection and dietary surveys were conducted for 36 students between April and May 2019. The study (including secondary use of dietary and urine data) was conducted in accordance with the principles of the Declaration of Helsinki, and all procedures were approved by the Nara Women’s University Ethics Committee (approval number 21–56) and registered to UMIN000052046 (https://center6.umin.ac.jp/cgi-open-bin/ctr_e/ctr_view.cgi?recptno=R000059406 accessed on 25 September 2024). Written informed consent was obtained from all participants before the use of the data acquired in this practical training and the analysis of collagen peptides in the 24 h urine specimen. Exclusion criteria were students who were taking collagen hydrolysate as supplements at the beginning of the study and students who had abnormal values as detected by regular medical check-ups at the university. The intake of supplement was confirmed from the results by weighed food record (WFR) survey. No participant met these exclusion criteria.

The 24 h urine collection was self-administered on the day of the WFR survey. Urinary specimens were collected using a portable urine measurement device (Urine Mate P; Sumitomo Bakelite, Tokyo, Japan), which can collect a 1/50 portion of all urine. The first urination after waking up in the morning was discarded, and the urinating time was noted as the start time of 24 h urine collection. Aliquots (1/50) of urine were collected as samples throughout the 24 h period from the time of the first urination. A few participants missed only one urine collection. In such cases, the 24 h urine volume was estimated from the average amounts of urine collected each time and the number of micturitions. After the collection, the urine was mixed with three volumes of ethanol; the supernatant was collected by centrifugation at 10,000× *g* for 5 min and stored at −30 °C until analysis.

Self-reported body height and weight were used to calculate the body mass index (BMI). Participants self-reported the type and duration of daily and weekly physical activities. Physical activity level (PAL) was determined using an add-in software of the Food Frequency Questionnaire, based on food groups (FFQg pack ver. 5, Kenpakusha, Tokyo, Japan) based on the physical data.

Food habits for the last few months were evaluated using the FFQ survey. Participants were asked about the amount and frequency of consumption of 29 food items and 10 kinds of cooked meals using a self-administered questionnaire provided with the FFQg pack. Intakes of food items and nutrients were calculated using the software based on the Standard Tables of Food Composition in Japan-2015 (STFC-2015) [24]. All food consumed in one whole day was evaluated by WFR. Participants were instructed by a dietician to record all food and beverages consumed in one whole day on weekdays. Food weights were measured by each participant. The intake of nutrients in the consumed foods was calculated based on the STFC-2015. The nutrients in the foods not listed in the STFC-2015 were estimated based on similar foods or food combinations listed in the STFC 2015.

### 2.3. Human Trial 2

This study was performed in accordance with the principles of the Declaration of Helsinki under the supervision of medical doctors, and all procedures were approved by the Nara Women’s University Ethics Committee (approval number 21–29). Participants were recruited from students and staff of Nara Women’s University through poster and leaflet distribution. The volunteers (average age 24.0 ± 3.1 years, average body weight 52.5 ± 6.4 kg) were informed of the objective and the potential risks of this study. Exclusion criteria were persons who were taking collagen hydrolysate as supplements and persons who had abnormal values detected during regular medical check-ups at the university.

Raw fillets of Japanese eels (*Anguilla japonica*) farmed in Tokushima Prefecture were ordered online and delivered in a refrigerated condition on ice. Fillets were cooked by two traditional Japanese grill methods: the Kansai (west Japan) style, in which the fillets were grilled for 4 min as they were, then flipped over and grilled for 2.5 min, and finally grilled with seasoning for 2.5 min with flipping, and the Kanto (east Japan) style, in which the fillets were steamed for 15 min, kept in the steamer for another 10 min after turning off the steam, and finally grilled with seasonings for 3 min with flipping. The meat juice from the steamed eel was also collected as test food. The seasoning was made mainly from soy sauce and sugar, which was served with the eel. The core temperature was confirmed to be above 75 °C for both cooking methods. Grilled eel fillet with the seasoning of both the Kansai and Kanto styles is referred to as kabayaki in Japanese. As estimated by amino acid analysis, 200 g of the grilled eel fillets prepared in the Kansai and Kanto styles together with meat juices contained 13.0 ± 1.0 g of collagen.

Following 12 h of fasting, seven healthy female volunteers ingested one of the test foods (200 g). After the washout period (1 week), the participants received another test food or 13 g of collagen hydrolysate in 100 mL of tap water. Collagen hydrolysate from fish scales (Ixos gel HDL-50SP; average molecular weight, 4769 Da) was a kind gift from Nitta Gelatin (Osaka, Japan). Approximately 4 mL of venous blood was collected from the cubital vein before ingestion and 30, 60, and 120 min after ingestion. Serum was prepared and mixed with three volumes of ethanol. Then, the supernatant was collected by centrifugation at 10,000× *g* for 5 min and stored at −30 °C until use.

### 2.4. Mouse Skin Culture

All experiments were conducted according to the ethical guidelines of the Kyoto University Animal Research Committee. The protocol was approved by the Kyoto University Animal Research Committee (permission number: R5–68). A five-week-old male Balb/c mouse was purchased from Japan SLC (Shizuoka, Japan). The mouse was euthanized by deep isoflurane anesthesia and the skin was sterilized with 70% ethanol. The body skin was shaved with a razor and immediately dissected using scissors. The skin was rinsed with D-PBS to remove residual ethanol and was cut into square pieces (approximately 6–7 mm in width). Eight pieces were placed directly on a polystyrene culture dish (90 mm i.d.) and covered with 3.5 mL of DMEM containing 10 μg/mL gentamicin, 50 U/mL penicillin, 50 μg/mL streptomycin, and 5% FBS-FHP; they were then maintained in a humidified incubator at 37 °C under 5% CO_2_. The medium was changed every 2–3 days. The collected media were combined with three volumes of ethanol, and the supernatant was collected by centrifugation at 10,000× *g* for 5 min and stored at −30 °C until further use.

### 2.5. Determination of Collagen Peptides

Ten collagen peptides in human urine and blood serum and mouse skin media were quantified by liquid chromatography-tandem mass spectrometry (LC-MS/MS) using an LCMS 8040 and LC-20 binary gradient system (Shimadzu, Kyoto, Japan) in multiple reaction monitoring (MRM) mode. Eight hydroxyproline-containing peptides (Ala-Hyp, Gly-Pro-Hyp, Phe-Hyp, Leu-Hyp, Hyp-Gly, Ile-Hyp, Pro-Hyp-Gly, and Ser-Hyp-Gly) were synthesized using the Fmoc strategy on an automated solid phase peptide synthesizer (PSSM-8; Shimadzu) as described earlier [19]. Aliquots (20 µL) of the 75% ethanol-soluble fractions of urine, blood serum, and mouse skin medium were dried in 1.5 mL tubes. The residue was dissolved in 20 µL of Milli-Q water, and 60 µL of 50 mM sodium borate buffer pH 8.8 and 20 µL of 0.3% AccQ were added. The reaction mixture was kept at 50 °C for 10 min and then diluted with 100 μL of 5 mM sodium phosphate buffer (pH 7.5) containing 5% acetonitrile and clarified by filtration using a Cosmonice filter (4 mm i.d., 0.45 μm, Nacalai Tesque). Quantification of each collagen peptide after the derivatization with AccQ was carried out by LC-MS/MS, as reported in our previous study [19]. In brief, AccQ derivatives of synthetic peptides were used for optimization of MRM conditions by using a LabSolutions Version 5.65 (Shimadzu). AccQ peptides were separated on an Inertsil ODS-3 column (2.5 µm, 2.1 × 250 mm; GL Science, Tokyo, Japan) using a binary gradient elution with solvents A (0.1% formic acid, 5% acetonitrile) and B (0.1% formic acid, 80% acetonitrile) at 0.2 mL/min. The gradient program was as follows: 0–12 min, 0–50% B; 12–20 min, 50–100% B; 20–24 min, 100% B; and 24.01–30 min, 0% B.

### 2.6. Amino Acid Analyses

Cooked fillets were freeze-dried using a freeze dryer (FDU-1200; Eyela, Tokyo, Japan) and then powdered in a blender (WB-1; Osaka Chemical, Osaka, Japan). The powder (1000 mg) was hydrolyzed in vacuo with 1 mL of 6 M HCl in a vial with a valve (Pierce vial, 40 mL; Thermo Fisher Scientific, Waltham, MA, USA) at 150 °C for 1 h. The hydrolysate was made up to 20 mL with water. The amino acids in the hydrolysate were analyzed according to the method of Bidlingmeyer et al. [25]. The Hyp contents in the meats were converted to collagen contents by multiplying by the coefficient (collagen weight/Hyp weight), which was obtained from the amino acid composition of collagen for eel. Free Hyp contents in blood serum were quantified without hydrolyzation.

### 2.7. Statistical Analyses

All statistical analyses were performed using SPSS version 26.0 (IBM, Armonk, NY, USA). Correlations between dietary intake, PAL, and urinary collagen peptide levels were assessed using the Spearman correlation coefficient because some of the urinary collagen peptide levels were not normally distributed. The *p*-value < 0.05 was considered statistically significant.

## 3. Results

### 3.1. Urinary Collagen Peptides

Table 1 shows the levels of collagen peptides in 24 h urine specimens. All 10 collagen peptides were detected in the urine. The most abundant urinary collagen peptide was Pro-Hyp, accounting for 78% of total collagen peptides. Other collagen peptides were minor peptides accounting for less than 5% of the total. However, the minor collagen peptides correlated well with the major Pro-Hyp content (0.78 ≤ r ≤ 0.90), except for Hyp-Gly. Hyp-Gly level showed the weakest correlation to major collagen peptide Pro-Hyp (r = 0.42). Correlation coefficients among the minor collagen peptides except for Hyp-Gly exhibited higher correlation coefficients (0.68 ≤ r ≤ 0.96) than those between Hyp-Gly and other minor collagen peptides (0.28 ≤ r ≤ 0.61), suggesting that Hyp-Gly may be produced in a manner that is at least partially different from other collagen peptides.

Table 2 shows physical characteristics, estimated nutrition intake, and meat consumption (g/day) on the day of urine collection and for a few months prior to urine collection. Meat consumption included seafood meat. Although BMI, PAL, energy, and protein intake were within the normal range, they were lower than the average levels of Japanese people of the same age and gender [26]. There was no significant correlation between body weight, BMI, or PAL and urinary collagen peptide levels. Estimated nutrient intake and meat consumption in the months preceding urine collection showed no significant correlation with urinary collagen peptide concentrations. On the other hand, protein intake on the day of urine collection significantly correlated solely with urinary contents of Hyp-Gly. Meat consumption significantly correlated with urinary collagen peptides (0.35 ≤ r) except for Gly-Pro-Hyp. Interestingly, urinary Hyp-Gly (r = 0.72) rather than the major Pro-Hyp (r = 0.37) showed the highest correlation coefficient for meat consumption among the urinary collagen peptides, indicating that ingestion of collagen-containing meat contributes more to the generation and urinary excretion of Hyp-Gly than to the production of other collagen peptides.

### 3.2. Blood Collagen Peptide Levels After Ingestion of Collagen-Rich Foods

Figure 1 shows the contents of free Hyp and total collagen peptide in blood before and after ingestion of the collagen-rich foods, grilled eel fillets prepared in the Kansai and Kanto styles and collagen hydrolysate. All the foods consumed contained 13 g of collagen and its degradation products. Levels of free Hyp and collagen peptide in human serum significantly increased after ingestion of collagen-rich foods. After ingestion of the collagen hydrolysate, both free Hyp and collagen peptide levels were significantly higher than those after ingestion of the grilled eel fillets. No significant differences were observed in the blood free Hyp and collagen peptide contents between the two groups that consumed Kansai- and Kanto-style grilled eel fillet.

Figure 2 shows the contents of each collagen peptide in human blood before and after ingestion of the collagen-rich foods. All collagen peptides were detected even before ingestion following 10 h of fasting. Before ingestion, the blood serum level of Pro-Hyp (0.32 µM) was significantly higher than that of Hyp-Gly and other collagen peptides (*p* < 0.001). After the ingestion of the grilled eel fillet prepared in both styles, all collagen peptides significantly increased. Interestingly, Hyp-Gly, a minor collagen peptide before ingestion, increased to a similar extent as Pro-Hyp, while serum levels of other minor collagen peptides after the ingestion were far less than Pro-Hyp and Hyp-Gly. After ingestion of 13 g of the collagen hydrolysate, Pro-Hyp and Hyp-Gly increased in a similar manner compared to other minor collagen peptides, whereas after the ingestion of grilled eel fillet, there was an increase in the levels of all collagen peptides.

### 3.3. Collagen Peptides Released from Cultured Mouse Skin

Figure 3 shows the contents of Pro-Hyp and Hyp-Gly released from mouse skin in the media. Contents of both Pro-Hyp and Hyp-Gly in the second medium (3–5 d from the start of incubation) were higher than those in the first medium (1–3 d), indicating degradation of mouse skin by proteases during the 5 day incubation period. In both the first and second medium, Pro-Hyp levels were approximately 500–1000-fold higher than Hyp-Gly levels.

## 4. Discussion

Taga et al. [21] demonstrated that levels of >20 collagen-derived peptides increase significantly in the urine for a few hours after ingestion of collagen hydrolysate. Among them, Pro-Hyp level was the highest. The present study also demonstrates the highest elevation of Pro-Hyp level in 24 h urine. Unexpectedly, the minor collagen peptide Hyp-Gly and not the major collagen peptide Pro-Hyp showed the highest correlation coefficient with meat consumption in 24 h urine on the day of urine collection (Table 2). Urinary Pro-Hyp has been demonstrated to be generated by degradation of endogenous collagen in pathological conditions such as muscular dystrophy [27]. Even in healthy individuals following overnight fasting, Pro-Hyp is the major collagen peptide in urine and blood [21,22,27]. These facts indicate that collagen peptides in human urine contain endogenous and food-derived collagen peptides. Pro-Hyp and longer collagen peptides in urine have been used as biomarkers for some pathological conditions [20,27,28,29,30], but Hyp-Gly has not been explored for this purpose. Some animal studies have demonstrated that collagen peptides are generated at wound healing sites and in inflammatory tissues [16,17]. In such cases, Pro-Hyp and some minor collagen peptides such as Leu-Hyp, as well as Gly-Pro-Hyp, significantly increase, while no significant increase in Hyp-Gly is observed [31]. Unlike Pro-Hyp, only negligible amounts of Hyp-Gly were released from skin (Figure 3). In contrast, Hyp-Gly has been reported to increase in human blood and urine after ingestion of collagen hydrolysate or collagen-containing food [19,21,22], which is consistent with the present study (Figure 2). These facts suggest that Hyp-Gly is preferentially generated during digestion and absorption of collagen-rich foods rather than from endogenous collagen. Pro-Hyp, produced by endogenous collagen degradation, can weaken the correlation between 24 h urinary Pro-Hyp concentration and meat consumption, indicating that Hyp-Gly, rather than Pro-Hyp, in 24 h urine and other minor collagen peptides reflect meat consumption on the urine collection day. However, there was no significant correlation between meat consumption during the past few months and urinary collagen peptides (Table 2). Thus, the 24 h urinary Hyp-Gly can only estimate the meat consumption on the same day.

Urinary histidine-related compounds such as 1-methylhistidine and 3-methylhistidine, carnitine, and taurine have been identified as potential biomarkers for meat consumption [32,33]. However, the taurine and carnitine contents in meat vary greatly depending on the type of meat [34,35]. Furthermore, methylhistamines also arise through endogenous muscle breakdown [36]. The present study is the first to show that Hyp-Gly, one of the degradation products of collagen, is preferentially generated by ingestion of collagen-containing meats rather than by the breakdown of endogenous collagen in female subjects. Therefore, 24 h urinary Hyp-Gly can be a good biomarker for estimating meat consumption. Conversely, when predicting endogenous collagen degradation using urinary collagen peptides, if we can predict the amount of collagen peptides produced from diet using Hyp-Gly, we can improve the accuracy of the evaluation of endogenous collagen degradation.

It remains unclear why only negligible amounts of Hyp-Gly are generated by the degradation of endogenous collagen, whereas the amounts of Hyp-Gly produced by the consumption of collagen-rich meat are high. Intact collagen in the body can be cleaved by specific endopeptidases such as matrix metalloproteinase-1, 8, and 13, resulting in formation of fragments with a triple helix [37]. These fragments are then presumably denatured by body temperature and converted to random coiled structures. The denatured collagen fragments are further degraded by many endopeptidases. Ingested collagen peptides and gelatin are degraded by digestive endopeptidases such as pepsin, trypsin, and chymotrypsin and finally by intestinal mucosa exopeptidases. Both Pro-Hyp and Hyp-Gly are known to be resistant to exopeptidases in blood and intestinal mucosa [12,38]. Therefore, the difference between endogenous and dietary Hyp-Gly levels in blood and urine may be attributed to differences in the substrate specificity of endopeptidases rather than exopeptidase in internal tissues and the digestive tract.

A limitation of this study is that the participants were limited to female university students. To our best knowledge, there have been no studies on the effects of different attributes, such as gender, age, and activity level, on collagen peptide composition in blood and urine before and after the ingestion of the same collagen-containing food. Although the age of the subjects and the collagen-containing food used were different from those in this study, our previous study showed that the blood Hyp-Gly (0.92 ± 0.60 µM) of six healthy male volunteers (mean age 31 years) after an overnight fast was significantly lower than the value of Pro-Hyp [19], as in the results of this study, but was slightly but significantly higher than the Hyp-Gly (0.24 ± 0.60 µM) of the female volunteers (mean age 24 years) in this study. On the other hand, the blood Hyp-Gly concentration after ingestion of another fish meat containing approximately the same amount of collagen was almost the same between the two studies, which suggests that 24 h urinary Hyp-Gly may increase in men, as in the results of this study, when collagen-containing food is consumed. However, as the Hyp-Gly content after overnight fasting, which is considered to be endogenous, is higher in men, the correlation between collagen-containing meat intake and 24 h urinary Hyp-Gly may be weaker in men than in women. These results suggest that age, sex, exercise level, and dietary habits may affect the synthesis of endogenous and dietary collagen peptides. These points should be investigated in the future. Furthermore, the urinary samples were not collected after the single ingestion of collagen-containing food, the grilled eel filet, as it was difficult to collect urine at the same time as blood.

The advantage of this study is that it has been shown not only through epidemiological experiments but also through intervention trials that Hyp-Gly is produced by the ingestion of collagen-containing food. Furthermore, it has been shown that only negligible amounts of Hyp-Gly are produced by endogenous collagen degradation, using human blood after fasting and a culture medium from mouse skin.

## 5. Conclusions

This study demonstrates that 24 h urinary Hyp-Gly, rather than the major urinary collagen peptide Pro-Hyp, reflects meat consumption on the day of 24 h urine collection. These findings indicate that among collagen peptides, Hyp-Gly is more likely to be produced from dietary collagen rather than endogenous collagen. Thus, 24 h urinary Hyp-Gly could serve as a potential marker for health monitoring.

## Figures and Tables

**Figure 1 nutrients-16-03574-f001:**
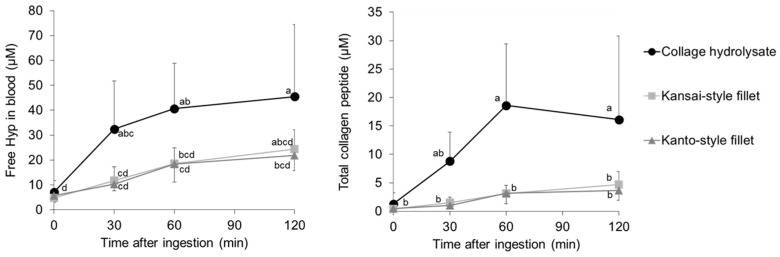
Free Hyp and total collagen peptides after ingestion of collagen-containing foods (n = 7). All symbols indicate average values. The different letters on the values indicate significant difference (*p* < 0.05).

**Figure 2 nutrients-16-03574-f002:**
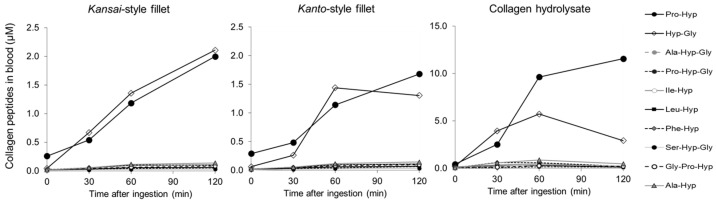
Content and structure of collage peptides in blood after ingestion of collagen-containing foods (n = 7). All symbols indicate average values.

**Figure 3 nutrients-16-03574-f003:**
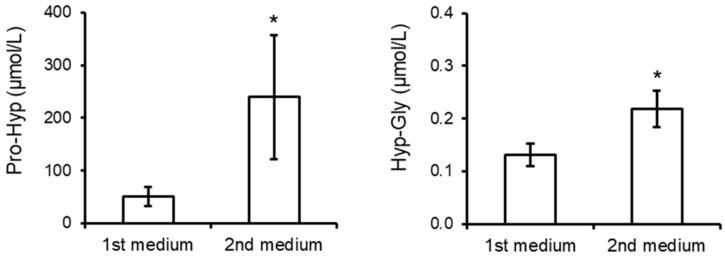
Contents of Pro-Hyp and Hyp-Gly in media with incubating mouse skin (n = 4). The data are shown as the mean ± SD. Asterisks indicate significant difference (*p* < 0.05).

**Table 1 nutrients-16-03574-t001:** Amounts of collagen peptides in 24 h urine and correlation coefficients between the peptides.

	nmol/24 h Urine		Spearman’s Correlation Coefficient									
	Median	(IQR)	Pro-Hyp	Ala-Hyp	Gly-Pro-Hyp	Phe-Hyp	Leu-Hyp	Hyp-Gly	Ile-Hyp	Pro-Hyp-Gly	Ala-Hyp-Gly	Ser-Hyp-Gly
Pro-Hyp	18,400	(6522–32,310)	1.00	**0.78**	**0.83**	**0.89**	**0.82**	**0.42**	**0.86**	**0.83**	**0.90**	**0.88**
Ala-Hyp	1136	(789–1830)		1.00	**0.83**	**0.89**	**0.94**	**0.45**	**0.93**	**0.78**	**0.87**	**0.82**
Gly-Pro-Hyp	856	(599–1219)			1.00	**0.87**	**0.83**	0.28	**0.80**	**0.68**	**0.79**	**0.71**
Phe-Hyp	840	(502–1,32)				1.00	**0.96**	**0.43**	**0.94**	**0.87**	**0.91**	**0.85**
Leu-Hyp	594	(391–801)					1.00	**0.51**	**0.96**	**0.87**	**0.87**	**0.81**
Hyp-Gly	514	(398–1215)						1.00	**0.61**	**0.61**	**0.49**	**0.48**
Ile-Hyp	483	(287–647)							1.00	**0.88**	**0.94**	**0.88**
Pro-Hyp-Gly	244	(144–284)								1.00	**0.85**	**0.84**
Ala-Hyp-Gly	235	(97–313)									1.00	**0.95**
Ser-Hyp-Gly	164	(73–312)										1.00

IQR, interquartile range. n = 35; r ≥ 0.34 *p* < 0.05, r ≥ 0.44 *p* < 0.01. Bold values indicate a statistically significant difference with a *p*-value less than 0.05.

**Table 2 nutrients-16-03574-t002:** Correlation coefficients of meat consumption and total protein and collagen peptides in 24 h urine.

		Spearman’s Correlation Coefficient									
	Median (IQR)	Pro-Hyp	Ala-Hyp	Gly-Pro-Hyp	Phe-Hyp	Leu-Hyp	Hyp-Gly	Ile-Hyp	Pro-Hyp-Gly	Ala-Hyp-Gly	Ser-Hyp-Gly
Physical characteristic											
Body weight (kg)	50.0 (45.3–54.5)	−0.01	0.09	0.09	0.12	0.15	0.15	0.08	0.09	0.02	−0.01
BMI (kg/m^2^)	19.7 (18.8–21.4)	0.16	0.19	0.16	0.02	0.23	0.10	0.26	0.18	0.14	0.16
PAL	1.57 (1.46–1.92)	0.17	0.29	0.20	0.27	0.31	0.23	0.27	0.21	0.19	0.16
Nutrient intake and meat consumption											
*-For a few months before urine collection*											
Energy (kcal/day)	1592 (1391–1870)	0.01	0.03	−0.05	−0.02	0.07	0.30	0.13	0.14	0.13	0.17
Protein (g/day)	54.2 (45.1–67.9)	−0.06	0.04	−0.02	−0.04	0.02	0.19	0.09	−0.01	0.08	0.02
Meat (g/day)	102.9 (65.7–125.7)	−0.07	0.06	−0.01	−0.05	0.03	0.14	0.07	0.01	0.01	−0.04
*-On the day of urine collection*											
Energy (kcal/day)	1418 (1189–1714)	0.06	0.10	0.01	0.03	0.077	0.24	0.16	0.09	0.17	0.15
Protein (g/day)	60.2 (40.7–71.4)	0.07	0.14	−0.04	0.10	0.18	**0.53**	0.26	0.31	0.22	0.27
Meat (g/day)	100 (24.2–156.0)	**0.41**	**0.46**	0.23	**0.39**	**0.49**	**0.75**	**0.55**	**0.64**	**0.47**	**0.48**

BMI, body mass index; PAL, physical activity level. IQR, interquartile range. n = 35; r ≥ 0.34 *p* < 0.05, r ≥ 0.44 *p* < 0.01. Bold values indicate a statistically significant difference with a *p*-value less than 0.05.

## Data Availability

All relevant data are available from the corresponding author on reasonable request.

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
