# Peer review of "Hydroxyprolyl-Glycine in 24 H Urine Shows Higher Correlation with Meat Consumption than Prolyl-Hydroxyproline, a Major Collagen Peptide in Urine and Blood"

_nutrients, 2024, doi:10.3390/nu16203574_

Round 1
Reviewer 1 Report
Comments and Suggestions for Authors
Collagen is a fibrous structural protein that is commonly found in animal tissues. Due to its tensile strength, it is a source of interest for many medical and cosmetic centres, and a properly selected diet and commonly available supplements are credited with a number of beneficial effects.
The introduction to the research presented in the article was written based on contemporary scientific literature, which was selected appropriately and logically.
The research was planned and conducted properly, and the methodology used was supported in many places by relevant literature items.
The description of the methods used may be surprising. While the authors have described in great detail the method of preparing eel fillets, the description of the HPLC-MS method used for quantitative analysis of peptides in blood and urine leaves much to be desired. There is no information about the column used, the flow rate of the eluent or its composition.
Figure 2 (line 290) is missing, so it is difficult to comment on the correctness of its description.
It is hard to disagree with the authors that the biggest limitation of the results they obtained (line 345) is the participation of only female students in the study. This is probably due to the place where the research was conducted (a women's university). Since the authors themselves note that eating habits, and consequently the content of appropriate peptides in urine, depend, among other things, on gender, it may have been worth trying to expand the study group to include male students. Do the authors have any evidence regarding the differences in the composition of the peptides studied in the blood and urine of men and women? It would certainly be necessary to expand the conducted research to obtain a fuller picture.
Author Response
Comments 1: The description of the methods used may be surprising. While the authors have described in great detail the method of preparing eel fillets, the description of the HPLC-MS method used for quantitative analysis of peptides in blood and urine leaves much to be desired. There is no information about the column used, the flow rate of the eluent or its composition.
Responses 1: Thank you for your valuable comments. We added the reference (19) for the previous study and the conditions of LC-MS/MS analyses (L216-222).
Comments 2: Figure 2 (line 290) is missing, so it is difficult to comment on the correctness of its description.
Responses 2: Figure 2 was present in the docx file used for first submission. Probably the Figure 2 might be removed during conversion to PDF file. I checked it appear correctly in the revised PDF.
Comments 3: It is hard to disagree with the authors that the biggest limitation of the results they obtained (line 345) is the participation of only female students in the study. This is probably due to the place where the research was conducted (a women's university). Since the authors themselves note that eating habits, and consequently the content of appropriate peptides in urine, depend, among other things, on gender, it may have been worth trying to expand the study group to include male students. Do the authors have any evidence regarding the differences in the composition of the peptides studied in the blood and urine of men and women? It would certainly be necessary to expand the conducted research to obtain a fuller picture.
Responses 3: Thank you for your valuable comment. I agree that this point is important. This study is the first to show that Hyp-Gly is preferentially generated by ingestion of collagen-containing foods rather than by the breakdown of endogenous collagen in female subjects. To the best of our knowledge, there are no studies on the effects of gender, age, and physical activity on the generation of dietary and endogenously derived collagen peptides in humans. However, in our previous study, although the age and collagen-containing foods used were different, the collagen composition in the blood was measured when male subjects ingested collagen-containing foods (fish). The results of the Hyp-Gly and Pro-Hyp contents before and after ingestion of collagen-containing foods in that study were almost the same as those of this study, but the Hyp-Gly content before ingestion was significantly higher than the results of this study. We cited these results and considered the possibility that there may be at least an influence of gender. These points are described in (L379- L397).
Reviewer 2 Report
Comments and Suggestions for Authors
The article provides interesting insights into collagen peptide markers for dietary meat intake. However, the introduction needs a clearer focus and the results require better interpretation. Addressing these issues would significantly enhance the readability and impact of the research. My suggestion are listed below:
The abstract lacks specific quantitative results, which makes it less informative for someone skimming the article to understand the key outcomes. Include specific results in the abstract, such as: "24-h urinary Hyp-Gly showed a correlation coefficient of r=0.72 with meat consumption, significantly higher than the coefficient for Pro-Hyp (r=0.37)." This makes the abstract more impactful and informative.
Line 87-90 – should not be in the introduction of the paper
Tables 1 and 2 are presented with minimal interpretation, which makes it difficult to understand the significance of the findings, particularly the correlation data. Correlation coefficients are given, but their implications are not well discussed. Please include a brief explanation of what each significant result means directly after each table.
For example, mention why the high correlation of Hyp-Gly with meat intake is important for dietary monitoring.
The authors mentions the limitation to female participants but doesn’t expand on how this affects the generalizability of the findings.
The authors should add a brief discussion on how the narrow demographic scope (only female students) limits the study's applicability to broader populations and suggest future studies on diverse groups.
The discussion does not adequately compare the findings of this study with those of similar previous research. Such comparisons would help contextualize the significance of this study and validate its findings, please integrate comparisons with previous literature more thoroughly.
The conclusion mentions that Hyp-Gly could be a marker for meat consumption but fails to detail how this could be practically used. Provide a practical context, such as suggesting the use of Hyp-Gly in dietary assessments for nutritional research or health monitoring.
Author Response
Comments 1: The abstract lacks specific quantitative results, which makes it less informative for someone skimming the article to understand the key outcomes. Include specific results in the abstract, such as: "24-h urinary Hyp-Gly showed a correlation coefficient of r=0.72 with meat consumption, significantly higher than the coefficient for Pro-Hyp (r=0.37)." This makes the abstract more impactful and informative.
Responses 1: Thank you for your valuable comment. We have revised the sentence in according to your comment (L23-26).
Comments 2: Line 87-90 – should not be in the introduction of the paper
Responses 2: We have deleted the sentence on line 87-90.
Comments 3: Tables 1 and 2 are presented with minimal interpretation, which makes it difficult to understand the significance of the findings, particularly the correlation data. Correlation coefficients are given, but their implications are not well discussed. Please include a brief explanation of what each significant result means directly after each table. For example, mention why the high correlation of Hyp-Gly with meat intake is important for dietary monitoring.
Responses 3: Thank you very much for your valuable comments. I added some sentences in according to your suggestions. Regarding Table 1, percentage of each peptide was added (L243-247. I also added a perspective that can be considered from the fact that only Hyp-Gly content had a low correlation with contents of the major component Pro-Hyp and other minor collagen peptides (L251-252). Regarding Table 2, I added the statement that although physical characteristics are within the normal range for Japanese people, these values are lower than the average (L261-263). I also added “ingestion of collagen-containing meat contributes more to the generation and urinary excretion of Hyp-Gly than to the production of other collagen peptides” (271-273).
Comments 4: The authors mention the limitation to female participants but doesn’t expand on how this affects the generalizability of the findings. The authors should add a brief discussion on how the narrow demographic scope (only female students) limits the study's applicability to broader populations and suggest future studies on diverse groups.
Response 4: Thank you for your valuable comment. I agree that this point is important. This study is the first to show that Hyp-Gly is preferentially generated by ingestion of collagen-containing foods rather than by the breakdown of endogenous collagen in female subjects. To the best of our knowledge, there are no studies on the effects of gender, age, and physical activity on the generation of dietary and endogenously derived collagen peptides in humans. However, in our previous study, although the age and collagen-containing foods used were different, the collagen composition in the blood was measured when male subjects ingested collagen-containing foods (fish). The results of the Hyp-Gly and Pro-Hyp contents before and after ingestion of collagen-containing foods in that study were almost the same as those of this study, but the Hyp-Gly content before ingestion was significantly higher than the results of this study. We cited these results and considered the possibility that there may be at least an influence of gender. These points are described in (L379- L397).
Comments 5: The discussion does not adequately compare the findings of this study with those of similar previous research. Such comparisons would help contextualize the significance of this study and validate its findings, please integrate comparisons with previous literature more thoroughly.
Responses 5: Thank you for your invaluable comments. I cited previously reported papers on biomarkers of meat intake and explained what is new about our findings (L353-364), and we cited our study using male subjects and add our previously mentioned discussion (L379-397).
Comments 6: The conclusion mentions that Hyp-Gly could be a marker for meat consumption but fails to detail how this could be practically used. Provide a practical context, such as suggesting the use of Hyp-Gly in dietary assessments for nutritional research or health monitoring.
Responses 6: Thank you for pointing that out. This study suggests the possibility of estimating endogenous and dietary collagen peptides separately using 24-hour urine samples, and I added the potential benefits of this (L360-364).
Round 2
Reviewer 2 Report
Comments and Suggestions for Authors
The authors have made significant improvements, I think the article can be published.